# Automatic Segmentation and Assessment of Valvular Regurgitations with Color Doppler Echocardiography Images: A VABC-UNet-Based Framework

**DOI:** 10.3390/bioengineering10111319

**Published:** 2023-11-16

**Authors:** Jun Huang, Aiyue Huang, Ruqin Xu, Musheng Wu, Peng Wang, Qing Wang

**Affiliations:** 1School of Biomedical Engineering, Southern Medical University, Guangzhou 510515, China; jwi11@sina.com (J.H.); huangaiyue1101@163.com (A.H.); musheng0624@gmail.com (M.W.); 13060791909@163.com (P.W.); 2The Second Affiliated Hospital of Guangzhou Medical University, Guangzhou 510260, China; xuruqin1989@126.com; 3Guangdong Provincial Key Laboratory of Medical Image Processing, Southern Medical University, Guangzhou 510515, China; 4Guangdong Province Engineering Laboratory for Medical Imaging and Diagnostic Technology, Southern Medical University, Guangzhou 510515, China

**Keywords:** valvular regurgitation, deep learning, automatic segmentation, automatic assessment, color Doppler echocardiography, valvular heart disease

## Abstract

This study investigated the automatic segmentation and classification of mitral regurgitation (MR) and tricuspid regurgitation (TR) using a deep learning-based method, aiming to improve the efficiency and accuracy of diagnosis of valvular regurgitations. A VABC-UNet model was proposed consisting of VGG16 encoder, U-Net decoder, batch normalization, attention block and deepened convolution layer based on the U-Net backbone. Then, a VABC-UNet-based assessment framework was established for automatic segmentation, classification, and evaluation of valvular regurgitations. A total of 315 color Doppler echocardiography images of MR and/or TR in an apical four-chamber view were collected, including 35 images in the test dataset and 280 images in the training dataset. In comparison with the classic U-Net and VGG16-UNet models, the segmentation performance of the VABC-UNet model was evaluated via four metrics: Dice, Jaccard, Precision, and Recall. According to the features of regurgitation jet and atrium, the regurgitation could automatically be classified into MR or TR, and evaluated to mild, moderate, moderate–severe, or severe grade by the framework. The results show that the VABC-UNet model has a superior performance in the segmentation of valvular regurgitation jets and atria to the other two models and consequently a higher accuracy of classification and evaluation. There were fewer pseudo- and over-segmentations by the VABC-UNet model and the values of the metrics significantly improved (*p* < 0.05). The proposed VABC-UNet-based framework achieves automatic segmentation, classification, and evaluation of MR and TR, having potential to assist radiologists in clinical decision making of the regurgitations in valvular heart diseases.

## 1. Introduction

Valvular heart disease (VHD) is a heart condition caused by structural abnormalities of the valves resulting from inflammation, myxomatous degeneration, degenerative changes, congenital malformations, ischemia, trauma, and other factors. Valvular regurgitation, one of common VHDs refers to the condition where the valves fail to close completely, causing some blood to flow back. Its impact on patients ranges from mild to severe. The severe regurgitations can impair cardiac functions and lead to heart failure. Therefore, early diagnosis, follow-up, and early intervention in valvular regurgitation are of great significance.

### 1.1. Assessment of Valvular Regurgitation

Transthoracic echocardiography (TTE) providing color Doppler echocardiography images is the most important imaging method for valvular regurgitation diagnosis and evaluation due to its widespread availability, non-invasiveness, low cost, acceptability, and safety profile [1]. TTE color Doppler echocardiography images provides information of cardiac structure, function, and hemodynamics and has a high sensitivity and specificity in assessing the flow of blood through the heart valves. Traditional methods for assessing the severity of regurgitation heavily rely on the subjective experience of cardiac radiologists, who select the peak regurgitation image from a sequence of color Doppler echocardiography images, manually outline one or more regurgitants and the atrium, and then evaluate the severity of regurgitation based on the extracted parameters. If the peak regurgitation image and the outline of the regurgitants and atrium are improperly selected, the manual tracing and calculation have to be repeated. Therefore, the repetitive manual operations are cumbersome.

### 1.2. Traditional Segmentation Methods

Generally, the segmentation methods of traditional medical ultrasound images are divided into threshold-based, clustering-based, watershed-based, graph-based, active contour model-based, and Markov random field-based segmentation methods [2]. The threshold-based segmentation methods are simple and effective, dividing the image into various regions with obvious contrast [3]. Santos et al. [4] used the Otsu threshold method to adaptively adjust the threshold value of left ventricular long-axis ultrasound images to extract the endocardial contour. However, the complex images cannot be effectively processed via the threshold-based segmentation methods, which are sensitive to mild changes in noise.

The cluster-based segmentation methods are image segmentation techniques based on pattern recognition. Those methods do not need boundary parameters, and self-adaptive changes can be achieved. Abdel-Dayem et al. [5] used the fuzzy C-means clustering algorithm to segment carotid arteries, resulting in more robustness to image noise and better resolution of fuzzy boundaries in ultrasound images. However, the susceptibility of the clustering process to external influences such as noise and outliers may lead to deviations. 

The watershed-based segmentation method is a graphical segmentation method based on image enhancement technology to achieve boundary segmentation of objects in an image. The continuous update of the grayscale values in the image achieves an accurate boundary segmentation. Gomez et al. [6] used watershed segmentation technology to effectively segment the diseased tissues in breast ultrasound images. However, the segmentation results obtained via the watershed-based segmentation method are unstable. 

The graph-based segmentation method uses the maximum flow in graph theory to solve image segmentation problems. It is versatile and flexible, and can be used to solve various image segmentation problems of different sizes and complexity levels. Chang-Ming et al. [7] proposed a graph cut model based on normalized segmentation criteria to segment cervical lymph nodes, which can effectively handle the speckle noise and low-contrast problems in ultrasound images. However, the graph segmentation technique is a search technique. If the search space is large, this method is time-consuming and slow. 

The segmentation method based on active contour model is based on graphical modeling and can segment objects in real time according to geometric shape and grayscale value. This method can effectively handle complex images and eliminate noise and interference. Hamou et al. [8] proposed a parametric active contour model to extract the left ventricle-endocardium contour. Dietenbeck et al. [9] and Alessandrini et al. [10] used the level set method to extract the myocardial contour and effectively address the problem of weak edges in the ultrasound image. However, the vulnerability of the method to noise prevents the model from extracting accurate contours and affect segmentation performance.

The segmentation method based on Markov random field (MRF) is a segmentation method based on probability and graph models. MRF allows the images with internal structures to have a precise segmentation without requiring excessive information. Xian et al. [11] proposed the MAP-MRF model, a Markov random field method based on maximum posteriori probability, to segment breast images for lesion tissue in low-quality ultrasound images. However, the training and testing time of the MRF-based method is very long and the method is limited to the segmentation of low-dimensional images. 

An early study applied a clustering-based method to segment aortic regurgitation (AR) in the Doppler echocardiography image [12]. However, there are still fewer applications of the aforementioned traditional segmentation methods in segmenting the valvular regurgitations.

### 1.3. Current Research on Evaluating Valvular Regurgitation

With the rapid development of artificial intelligence, deep learning-based methods have been widely used in medical image processing. The deep learning segmentation network based on convolutional neural networks is able to restore the category of each pixel from the abstract features, transform the image-level segmentation problem into a classification decision problem [13], achieve pixel-level classification [14], and then improve the segmentation accuracy and efficiency. Convolutional neural network (CNN) is one of the representative algorithms of deep learning [15]. Based on the CNN model, some studies proposed V-Net [16], E-Net [17], Omega-Net [18] and RDNet [19], and successfully applied them to cardiac ventricle segmentation. These models can automatically learn complex image features and concepts without the need for manual feature engineering or prior knowledge encoding. Ge et al. [20] improved the U-Net model to achieve automatic segmentation of the left ventricle in echocardiographic images. These studies contributed to automatic segmentation of cardiac ventricles. Recently, Zhang et al. [21] applied deep learning to benefit the diagnosis of VHD and proposed a Mask R-CNN model for automatically assessing the severity of mitral regurgitation (MR). However, that study was limited to a single MR without other regurgitations such as tricuspid regurgitation (TR). Moreover, the relatively low accuracy of regurgitation assessment was needed to be improved.

### 1.4. The Study Proposed in This Article to Evaluate Valve Regurgitation

Herein, this study aims to investigate automatic segmentation, classification, and evaluation of valvular regurgitations using a deep learning-based method with color Doppler echocardiography images. A VABC-UNet model was proposed consisting of VGG16 encoder, U-Net decoder, batch normalization, attention block and deepened convolution layer based on the U-Net backbone. Then, a VABC-UNet-based framework was established for automatic segmentation, classification, and evaluation of valvular regurgitations. The performance of the VABC-UNet model was evaluated and compared with U-Net and VGG16-UNet. The hypothesis of this study is that the proposed VABC-UNet-based framework achieves a better performance than U-Net and VGG16-UNet in segmentation, classification, and evaluation of MR and TR. The scientific significancy of this study is that the proposed framework has the potential to improve the automatically diagnostic efficiency and accuracy of valvular regurgitation and assist radiologists in clinical decision making of the regurgitations in VHDs.

## 2. Materials and Methods

The research framework of this study is shown in Figure 1. The color Doppler echocardiography images of the heart with MR or TR in an apical four-chamber view were selected in this study. After marking the regurgitant jet and atrium, the training and testing sets were established. The proposed VABC-UNet-based framework was established for automatic segmentation, classification, and evaluation of valvular regurgitations. The performance of the VABC-Unet model was evaluated in comparison with the classic U-Net and VGG16-Unet network models. 

### 2.1. Image Acquisition and Preprocessing

TTE was performed by a cardiac radiologist with experience in color Doppler echocardiography using an ultrasound diagnostic system (EPIQ 7C, Philips Medical Systems, Bothell, WA, USA). The color Doppler echocardiography images of the heart in an apical four-chamber view were acquired. A total of 315 images of MR and/or TR were collected from December 2020 to June 2021. This study was approved by the Institutional Ethics Committee of the Second Affiliated Hospital of Guangzhou Medical University.

The original frame images contained irrelevant information such as patient information and acquisition parameters that were not relevant to the valve regurgitation. To minimize the impact of such information on the subsequent model learning, the original frame images were cropped to only retain the target region. The image size was cropped from 600 × 800 to 480 × 576.

### 2.2. Image Annotation

Under the guidance of cardiologists, the processed frame images were annotated. The LabelMe annotation tool [22] was used to demarcate the region of interest (RoI) including the regurgitation area and corresponding atrial contour in each frame.

### 2.3. Dataset Establishment

The annotated images and their corresponding mask images were then divided into training and test datasets at a ratio of 8:1. Specifically, there are 280 images including 139 MR images and 141 TR images in the training set for training the models, while 35 images were included in the testing set including 21 MR images and 14 TR images for evaluating and validating the effectiveness of the model.

Considering the need for a large amount of data for training the model, this study employed data augmentation techniques including mirroring, rotation, scale transformation, and grayscale transformation to increase the number of the original and mask images in the training dataset. The augmented training set consisted of 3920 images. Data augmentation improves the sample diversity of the training set and makes the model learn the image features with additive noise and geometric deformation characteristics during training. Consequently, overfitting can be prevented, and the model’s robustness and stability are improved.

### 2.4. Establishment of VABC-UNet Model for Segmentation

The VGG model is a CNN architecture [23]. Our recent related work has proposed an improved VGG16-Unet model [24] with strong feature extraction capacity, which adopts the U-Net structure, but the encoder is replaced with the VGG16 encoder. The VGG16 encoder was modified with retaining only the first three fully connected layers and adding batch normalization (BN) and a deepened convolution layer in each encoding block. Here BN helps to avoid neuron saturation at some extent and improve convergence speed, learning efficiency, and stability of the model [25].

Furthermore, based on our recent work proposing an attention-VGG16-Unet model [26], we added the attention block [27] to the U-Uet decoder to prevent pixel-level information loss and improve the accuracy of feature extraction and segmentation of target tissues in ultrasound images. The attention block performs two steps as follows. First, the RoI (attention focus) is selected after a quick scan of the entire image. Then, a stronger attention is assigned to the focused regions. The schematics of the attention mechanism are shown in Figure 2 The feature vectors extracted from the encoder and decoder are separately input the attention block, passed through a fully connected layer, and then added together. The sum result passes through a ReLU activation function and another fully connected layer. Afterwards, a Sigmoid activation function is applied to obtain the attention weights. Those weights are element-wise multiplied with the feature vectors extracted from the decoder. Finally, the output of a feature map of the same type as the input is achieved. In this study, the attention block was combined with skip connection in the proposed VABC-Unet model.

Meanwhile, an extra convolutional layer [28] is added to each encoding block to achieve a better feature extraction capacity and more efficiently extracted high-level semantic information in this study.

According to the aforementioned modifications, the modified model based on VGG16-Unet was named as VABC-Unet in this study and used for the segmentation of the regurgitation and atrium. The Loss function of the model was defined as the cross-entropy function. Figure 3 shows the network architecture of the VABC-Unet model.

### 2.5. Model Performance Evaluation

This study evaluated the performance of the proposed VABC-Unet model in comparison with the classic U-Net and VGG16-Unet models. Five evaluation metrics, Dice, Jaccard, Precision, and Recall are used to quantitatively evaluate the performance of the models in segmenting the regurgitation and atrium. The manually segmented annotation results are the ground truth (GT) represented by $\Omega_{GT}$. Equations (1)–(4) are the definitions of the five metrics. Dice and Jaccard measure the spatial overlap between the segmented region by the model and the GT, while Precision calculates the ratio of positive pixels in the segmentation results. Recall is defined as the proportion of correctly segmented pixels in the GT.
(1)$$D\mathrm{ice}=\frac{2\cdot \left|\Omega_{GT}\cap \Omega_{SC}\right|}{\left|\Omega_{GT}\right|+\left|\Omega_{SC}\right|}=\frac{2\cdot TP}{2\cdot TP+FP+FN}$$
(2)$$J\mathrm{accard}=\frac{\left|\Omega_{GT}\cap \Omega_{SC}\right|}{\left|\Omega_{GT}\cup \Omega_{SC}\right|}=\frac{TP}{TP+FP+FN}$$
(3)$$P\mathrm{recision}=\frac{\left|\Omega_{GT}\cap \Omega_{SC}\right|}{\Omega_{SC}}=\frac{TP}{TP+FP}$$
(4)ReCall=|ΩGT∩ΩSC|ΩGT=TPTP+FN
where $\Omega_{SC}$ represents the segmented region obtained by the model, and TP, FP, and FN represent the pixel numbers of true positive, false positive, and false negative, respectively.

### 2.6. Classification of Regurgitation

Due to the irregular shape and location of the automatically segmented apical region, this study classified the regurgitations by positioning the centroid of the apical region. The features of the regurgitation jet and atrium extracted by the VABC-Unet in the previous section were input into the classification module. Through locating the centroid coordinates of the atrial region, the position of the atrium in the color Doppler echocardiography image in the four-chamber view is determined, and consequently, the regurgitation can be automatically classified (Figure 4).

The centroid of a planar area can be obtained by taking a weighted average of the target coordinate values, as shown in Equations (5) and (6).
(5)$$\bar{x}=\frac{\sum_{i=1}^{n}x_{i}p(x_{i},y_{i})}{\sum_{i=1}^{n}p(x_{i},y_{i})}$$
(6)$$\bar{y}=\frac{\sum_{i=1}^{n}y_{i}p(x_{i},y_{i})}{\sum_{i=1}^{n}p(x_{i},y_{i})}$$
where $\bar{x}$ and $\bar{y}$ represent the centroid coordinates of the target, n represents the number of the pixels in the target region, $x_{i}$ and $y_{i}$ represent the coordinates of the *i*-th pixel, and $p(x_{i},y_{i})$ is the grayscale value of the *i*-th pixel. When the y-coordinate of the centroid is less than half of the image height, the regurgitation is TR if the centroid x-coordinate is on the left side of the image and belongs to the right atrium; otherwise, the regurgitation is MR. If there are two centroids located at the left and right sides of the image, respectively, the regurgitation is a hybrid of TR and MR. Meanwhile, the module determines that the regurgitation is multiple or single.

### 2.7. Evaluation of Regurgitation Severity

According to the 2021 expert consensus on standardized echocardiographic evaluation of valvular regurgitation in the adult Chinese population [29] and the current commonly used fast grading criteria [30], the ratio of the valvular regurgitation jet area to the corresponding atrium area is defined as a semi-quantitative indicator to assess the severity of regurgitation. As shown in Figure 5, the severity of MR is evaluated into three grades (mild, moderate, and severe), while the severity of TR is evaluated into four grades (mild, moderate, moderate–severe, and severe). The automatic evaluation results of the regurgitation severity are compared with those by manual annotation to analyze the accuracy of the evaluation by the models.

### 2.8. Software and Hardware Environment for Model Operation

In this study, the operating environment is a server with an Intel(R) Core(TM) i7-6700 CPU@3.40GHz, 8GB RAM, and NVIDIA TITAN X graphics card. The network architecture was constructed in Python and TensorFlow framework using the Keras deep learning library.

## 3. Results

### 3.1. Learning Ability of the Models during Training

During the training process of segmentation of the regurgitant jet and atrium, the performance of the models was evaluated with an increasing number of Epochs. In Figure 6, as the Epochs number increases, the accuracy, Dice, and Jaccard of the models improve, while the Loss decreases. Moreover, compared the metric change curves of the VABC-Unet model with other two models, it is evident that the VABC-Unet model outperforms the other two models.

Similar results of the atrium segmentation via the models are obtained in Figure 7. After validation, the Epochs was set to 50 in this study. It can be seen that the changes in the metrics slow down when Epochs > 25, and then the values change slightly when Epochs is close to 50. Therefore, this study selected Epochs = 50 for model training.

### 3.2. Comparative Results of Segmentation

Figure 8 shows the segmentation results of three cases by the proposed VABC-Unet, U-Net, and VGG16-Unet models in comparison with GT. The results indicated that better segmentations of regurgitant jet and atrium were obtained via the VABC-Unet model than those of the other two models. Figure 9 shows that the proposed VABC-Unet model segments fewer pseudo- and over-segmentations than the U-Net and VGG16-Unet models.

Table 1 and Table 2 lists the quantitative results of the Dice, Jaccard, Precision, and Recall for the segmentation of the regurgitation jet and atrium. The results are expressed as mean ± standard deviation (SD). Statistical significance is considered between two network models using pair *t*-test at *p* < 0.05. For the regurgitation segmentation, in comparison with the U-Net model, the Dice, Jaccard, and Precision values of the VGG16-Unet and VABC-Unet models have been improved significantly (*p* < 0.01). Compared with the VGG16-Unet model, the Recall value of the VABC-Unet model improved significantly (*p* < 0.01), while the Dice and Jaccard values of the VABC-Unet model insignificantly increase and the precision value insignificantly reduces. For atrium segmentation, a similar result is obtained in that using the VABC-Unet models, a better segmentation of the atrium could be achieved compared with the U-Net and VGG16-Unet. The Dice and Jaccard values of the VABC-Unet model significantly improve (*p* < 0.05) compared with the U-Net and VGG16-Unet, whereas the Recall and Precision values significantly improve (*p* < 0.01) in comparison with the U-Net and VGG16-Unet, respectively. As the whole, the VABC-Unet model shows the best performance of segmentation in comparison with the other two models.

### 3.3. Comparative Results of Regurgitation Classification

Using manual classification of valvular regurgitation as GT, the accuracy values of regurgitation classification via various models are listed in Table 3. It is found that compared to the accuracy of the U-Net model, the classification accuracy could be improved by the VGG16-Unet and further improvements in the classification accuracy by the VABC-Unet model are achieved. The overall classification accuracy of all the regurgitations in the test dataset by the U-Net, VGG16-Unet, and VABC-Unet was 0.74, 0.89, and 1, respectively.

### 3.4. Comparative Results of Regurgitation Severity Evaluation

Using manual assessment of regurgitation severity as GT, the accuracy values of regurgitation severity evaluated by various models are listed in Table 4. It is found that the mild regurgitations are evaluated with a lowest accuracy (0.48) using the U-Net model, but the moderate and severe regurgitations are easily evaluated (0.78 and 1, respectively). The evaluation performance of the VGG16-Unet is improved but it is noticed that the accuracy of evaluating the severe regurgitations is 0, which means that the VGG16-Unet model fails to provide a correct evaluation of 2 cases of severe regurgitation in the test set. The VABC-Unet model can obtain higher overall accuracy value (0.94) for all the regurgitations in the test dataset than that of the U-Net and VGG16-Unet (0.6 and 0.77, respectively). Figure 10 shows that only a few cases of mild and moderate regurgitations are misinterpreted by the proposed VABC-Unet model.

## 4. Discussion

### 4.1. Segmentation Performance of the VABC-Unet Model

The results of this study indicated that the U-Net and VGG16-Unet models segmented TR and MR with pseudo- and over-segmentation, and consequently, the severity evaluations were not satisfactory. Compared with the U-Net and VGG16-Unet models, the VABC-Unet model proposed in this study significantly improved the segmentation performance by reducing pseudo- and over-segmentation, and greatly increased the accuracy of valvular regurgitation assessment. The VABC-Unet model has made the following improvements to enhance its segmentation performance. First, the model adopts the basic architecture of the VGG16-Unet, which possesses strong feature extraction capabilities [24,26]. The deepened convolutions enhanced feature extraction capability of the model [28]. Secondly, the normalization operations enhanced convergence of the model [25]. Additionally, an attention mechanism added in the decoder optimized the extracted features [26].

### 4.2. Classification of the VABC-Unet-Based Framework

The classification performance of the framework depended on the features of regurgitation jet and atrium extracted by the VABC-Unet. The proposed VABC-Unet model effectively addressed the issues of pseudo- and over-segmentation, greatly improving the accuracy of valvular regurgitation classification. Previous studies focused on only one type of valvular regurgitation, for example, MR [31] and TR [32]. There is currently limited research on automatic classification of MR, TR, and other regurgitations. In this study, the proposed model could automatically classify the regurgitations (TR, MR, or hybrid) with a higher accuracy compared with the U-Net and VGG16-Unet models. More color Doppler echocardiography images of the aortic regurgitation (AR) and pulmonary regurgitation (PR) should be collected and involved in future work.

### 4.3. Severity Evaluation of the VABC-Unet-Based Framework

The research on applying deep learning methods for evaluating valvular regurgitation is currently paid more attention. Zhang et al. [21] established the Mask R-CNN model to evaluate the severity of MR, which achieved satisfactory results. However, the accuracy of moderate regurgitation assessment was low, equal to 0.81, mainly due to the partial overlap between the characteristics of grade III (moderate) and grade IV (severe) regurgitation, making it difficult to distinguish between moderate and severe regurgitation. Moreover, their research was limited to MR, and did not evaluate the severity of TR. This study established the VABC-Unet model to automatically assess the severity of the regurgitations with a higher accuracy of 0.94. Recently, Yang et al. [31] provided a self-supervised learning-assisted approach of MR severity grading, while Huang et al. [32] applied deep learning for TR severity grading. The results of those studies showed that improvements in more accurate evaluation of valvular regurgitation are still needed. In this study, the deep learning-based framework was established for automatic segmentation, classification, and evaluation of both MR and TR. Based on the improvements of feature extraction of the VABC-Unet, the evaluation accuracy for mild, moderate, moderate–severe, and severe regurgitation improved obviously.

In addition to the fast grading method [30], there are other methods for assessing the degree of valvular regurgitations, such as the proximal isovelocity surface area (PISA) method [33], which assesses the magnitude of the regurgitation flow using the effective regurgitant orifice area (EROA) and the regurgitant velocity time integral (VTI). There is a mild impact of the changes in hemodynamics of valvular regurgitations on the PISA assessment. Moreover, due to irregular areas of regurgitations, three-dimensional PISA can be used to compensate for the shortcomings of two-dimensional ultrasound. However, the accuracy of the PISA method is vulnerably affected by the measurement of the contour radius of the isokinetic hemisphere of eccentrically regurgitated blood flow, so the PISA method is not suitable for simultaneously occurred multiple regurgitations. In view of other influence factors and large inter-observer variability, the PISA is not used in this study.

The results of this study demonstrated that the proposed VABC-Unet-based framework had a good performance in segmentation, classification, and assessment of valvular regurgitation. However, there are still limitations that need further exploration in future work. First, the type and quantity of the valvular regurgitations included in this study were limited. The randomly selected test dataset included 23 cases of MR and 12 cases of TR, of which 23 regurgitations were mild, 9 moderate, 1 moderate–severe, and 2 severe. The number of moderate–severe and severe regurgitations was relatively small. The lack of comprehensiveness may lead to potential errors. Secondly, the generalization ability of the model still need to improve, although data augmentation techniques were used to increase the quantity and diversity of training samples. Thirdly, automatic assessment of all types of valvular regurgitations would be achieved using the proposed framework in future work.

## 5. Conclusions

This study established a deep learning-based framework for automatic segmentation, classification and evaluation of valvular regurgitations. A VABC-Unet model was proposed with improvements of the VGG16 encoder, attention module, normalization operations, and additional convolution operations based on the classic U-Net model. The proposed hypothesis is proved by this study. The results demonstrate that the VABC-Unet model has better segmentation of regurgitant jets and corresponding atrium with few pseudo- and over-segmentations in comparison with the U-Net and VGG16-Unet models. Consequently, the accuracy of classification and evaluation of the proposed VABC-Unet-based assessment framework improves greatly. This study indicated that the proposed framework has potential to improve the automatically diagnostic efficiency and accuracy of valvular regurgitation and assist radiologists in clinical decision making of the regurgitations in VHDs.

However, there are still some limitations of this study, such as the limited type of the valvular regurgitations and the limited number of the color Doppler echocardiography images of MR and/or TR in an apical four-chamber view. Additionally, the small data volume of the test set leads to some unsatisfactory segmentation results. Although data augmentation techniques were used to increase the number and diversity of training samples, the generalization ability of the model still needs to be improved. In the future, more patient-derived data will be included to expand the dataset to further improve the generalization ability of the model, and increase the accuracy of automatic evaluation of valvular regurgitation severity. In addition, pulmonary regurgitation and aortic regurgitation should also be included. The proposed VABC-UNet-based framework will achieve the automatic evaluation of all types of valve regurgitation. Future investigations will be also performed on the AI algorithms applied in evaluating the regurgitation via the PISA method.

## Figures and Tables

**Figure 1 bioengineering-10-01319-f001:**
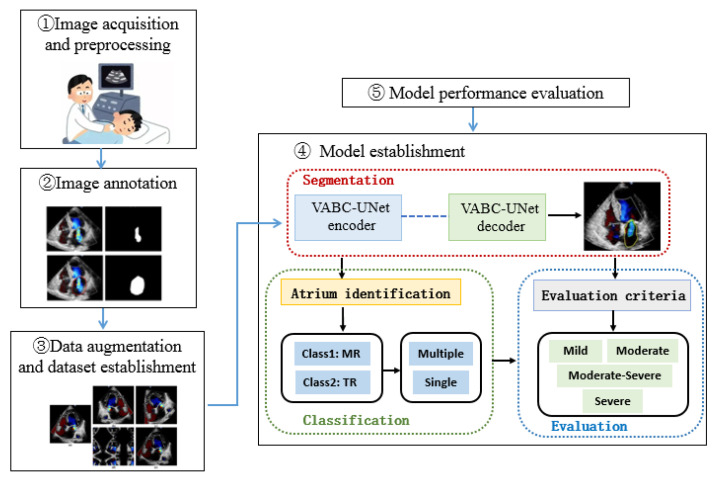
Schematic of the automatic assessment framework based on a VABC-Unet model for valvular regurgitation.

**Figure 2 bioengineering-10-01319-f002:**
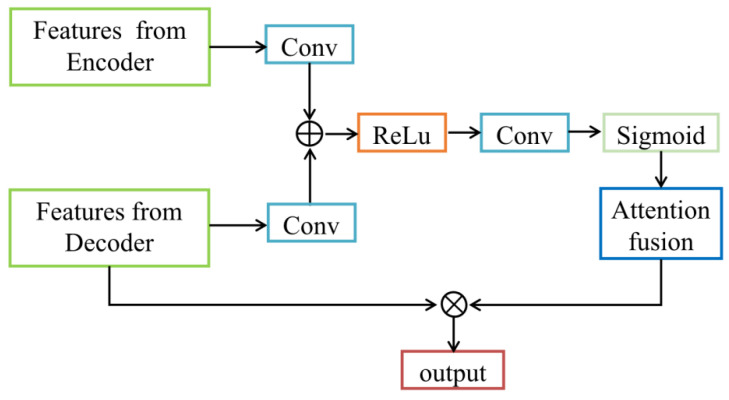
Schematic diagram of the attention block.

**Figure 3 bioengineering-10-01319-f003:**
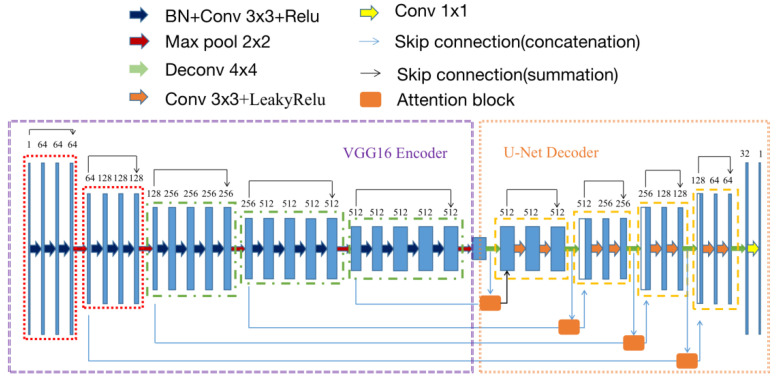
The network architecture of the proposed VABC-Unet model.

**Figure 4 bioengineering-10-01319-f004:**
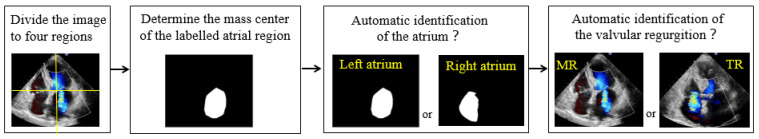
Schematics of automatic classification of MR and TR.

**Figure 5 bioengineering-10-01319-f005:**
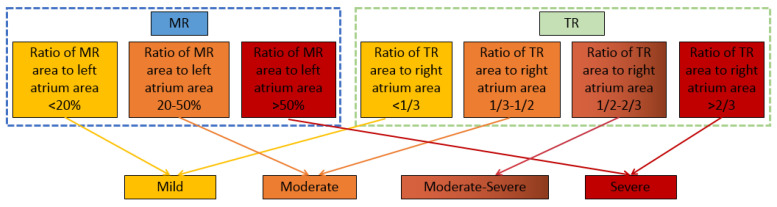
Evaluation criteria for severity of valvular regurgitation.

**Figure 6 bioengineering-10-01319-f006:**
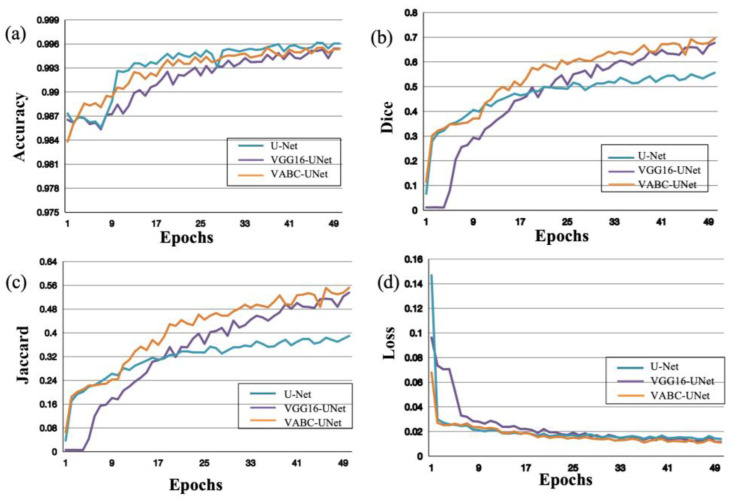
Performance of the proposed VABC-Unet model in comparison with the U-Net and VGG16-Unet models during regurgitation jet segmentation. (**a**–**d**) The curves of accuracy, Dice, Jaccard, and Loss with increase in Epochs.

**Figure 7 bioengineering-10-01319-f007:**
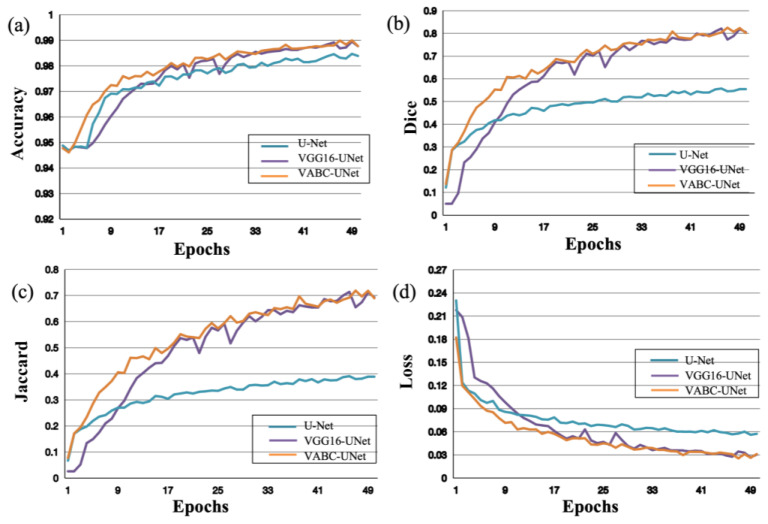
Performance of the proposed VABC-Unet model in comparison with the U-Net and VGG16-Unet models during atrium segmentation. (**a**–**d**) The curves of accuracy, Dice, Jaccard, and Loss with increase in Epochs.

**Figure 8 bioengineering-10-01319-f008:**
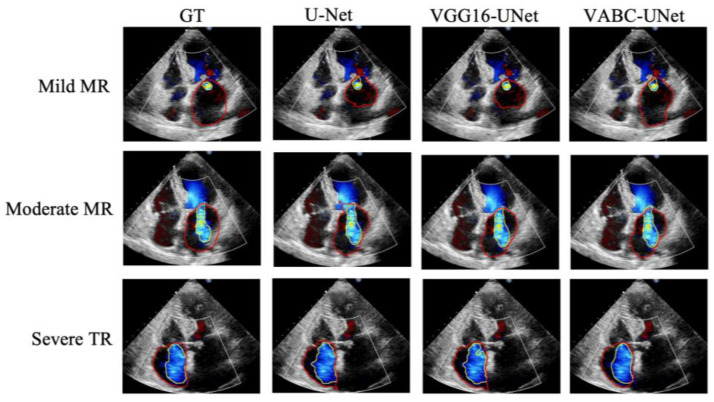
The segmentation results of three cases by the proposed VABC-Unet, U-Net, and VGG16-Unet models in comparison with GT. Yellow outlines represent the segmentations of the regurgitant jets. Red outlines represent the segmentations of the atriums.

**Figure 9 bioengineering-10-01319-f009:**
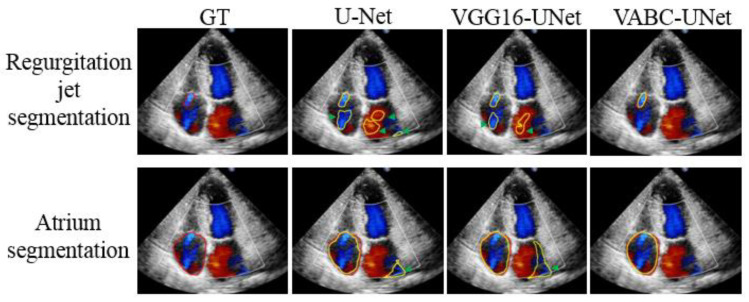
Pseudo- and over-segmentations of the regurgitation jets and atriums via the proposed VABC-Unet, U-Net, and VGG16-Unet models in comparison with GT. Red outlines represent GT. Yellow outlines represent the automatic segmentations. Green arrows show pseudo- and over-segmentations.

**Figure 10 bioengineering-10-01319-f010:**
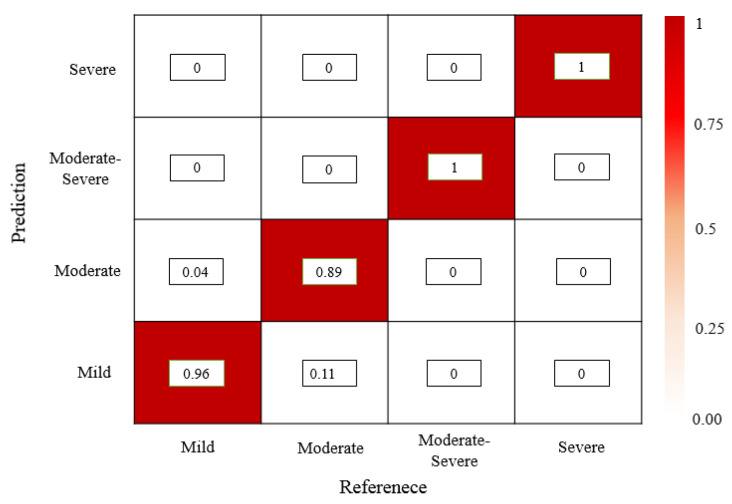
The confusion matrix of evaluation results of the regurgitation severity via the VABC-Unet model.

**Table 1 bioengineering-10-01319-t001:** Performance evaluation results (mean ± SD) of regurgitation segmentation.

Models	Regurgitation Segmentation			
	Dice	Jaccard	Precision	Recall
U-Net	0.718 ± 0.180	0.587 ± 0.199	0.626 ± 0.213	0.905 ± 0.049
VGG16-Unet	0.806 ± 0.048 **	0.678 ± 0.068 **	0.778 ± 0.087 **	0.848 ± 0.079 **
VABC-Unet	0.810 ± 0.098 **	0.690 ± 0.115 **	0.756 ± 0.122 **	0.890 ± 0.059 *^,‡^

* Statistically significant difference at *p* < 0.05 vs. U-Net; ** Statistically significant difference at *p* < 0.01 vs. U-Net; ^‡^ Statistically significant difference at *p* < 0.01 vs. VGG16-Unet.

**Table 2 bioengineering-10-01319-t002:** Performance evaluation results (mean ± SD) of atrium segmentation.

Models	Atrium Segmentation			
	Dice	Jaccard	Precision	Recall
U-Net	0.882 ± 0.079	0.797 ± 0.115	0.903 ± 0.081	0.878 ± 0.118
VGG16-Unet	0.904 ± 0.046	0.828 ± 0.072 **	0.860 ± 0.072 **	0.959 ± 0.044 **
VABC-Unet	0.915 ± 0.041 **^,†^	0.846 ± 0.064 **^,†^	0.907 ± 0.052 ^‡^	0.927 ± 0.063 **^,‡^

** Statistically significant difference at *p* < 0.01 vs. U-Net; ^†^ Statistically significant difference at *p* < 0.05 vs. VGG16-Unet; ^‡^ Statistically significant difference at *p* < 0.01 vs. VGG16-Unet.

**Table 3 bioengineering-10-01319-t003:** Classification results of regurgitation type.

Models	Accuracy		
	MR	TR	Overall
U-Net	0.87	0.5	0.74
VGG16-Unet	0.96	0.75	0.89
VABC-Unet	1	1	1

**Table 4 bioengineering-10-01319-t004:** Assessment results of regurgitation severity.

Models	Accuracy				
	Mild	Moderate	Moderate–Severe	Severe	Overall
U-Net	0.48	0.78	1	1	0.6
VGG16-Unet	0.83	0.78	1	0	0.77
VABC-Unet	0.96	0.89	1	1	0.94

## Data Availability

Currently, the datasets generated and analyzed during the current study cannot be made publicly accessible due to privacy protection.

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
