# Peer review of "Automatic Segmentation and Assessment of Valvular Regurgitations with Color Doppler Echocardiography Images: A VABC-UNet-Based Framework"

_bioengineering, 2023, doi:10.3390/bioengineering10111319_

Round 1
Reviewer 1 Report
Comments and Suggestions for Authors
Review comments on “Automatic Segmentation and Assessment of Valvular Regurgitations with Color Doppler Echocardiography Images: A VABC-UNet-Based Framework” by Jun Huang etc.
This work presents a VABC-UNet model for automatic segmentation and classification of mitral regurgitation (MR) and tricuspid regurgitation (TR) using a deep learning-based method, aiming to improve the efficiency and accuracy of diagnosis of valvular regurgitations.
My main general comments are as below:
- The authors didn’t provide a comparison of the performances on training and testing sets. The authors should investigate experimentally the overfitting of the proposed model.
- The authors use a small dataset. Authors should consider using pre-trained neural networks Vision Transformers (ViT) or U-net for automatic segmentation and classification of mitral regurgitation (MR) and tricuspid regurgitation (TR).
- The work will be significant if the source codes are presented to the public for a detailed analysis of the proposed method.
- The authors should share the dataset to be downloaded freely to the industry and research community.
- Conclusions need more elaboration about: outcomes, limitations, and possible/future scenarios.
- The authors should investigate the stability of the proposed method because image can be degraded by additive noise, in the presence of cluttering backgrounds, geometric modifications such as pose changing and scaling, nonuniform illumination, and eventual object occlusions.
Reviewer 2 Report
Comments and Suggestions for Authors
I consider that the manuscript can be published in the present form.
Author Response
Thank you very much for your reviewing our manuscript.
Reviewer 3 Report
Comments and Suggestions for Authors
The new version had improved substantially that merits publication in Bioengineering
Author Response

(The authors gave the same response as above.)

Round 2
Reviewer 1 Report
Comments and Suggestions for Authors
All suggestions and comments were corrected